# Identification of Homologous Polyprenols from Thermophilic Bacteria

**DOI:** 10.3390/microorganisms9061168

**Published:** 2021-05-28

**Authors:** Lucia Gharwalová, Andrea Palyzová, Helena Marešová, Irena Kolouchová, Lucie Kyselová, Tomáš Řezanka

**Affiliations:** 1Department of Biotechnology, Faculty of Food and Biochemical Technology, University of Chemistry and Technology Prague, 166 28 Prague, Czech Republic; lucia.gharwalova@gmail.com (L.G.); irena.kolouchova@vscht.cz (I.K.); 2Institute of Microbiology, The Czech Academy of Sciences, 142 20 Prague, Czech Republic; palyzova@biomed.cas.cz (A.P.); maresova@biomed.cas.cz (H.M.); 3Research Institute of Brewing and Malting, 120 44 Prague, Czech Republic; kyselova@beerresearch.cz

**Keywords:** polyprenols, thermophilic bacteria, *Geobacillus stearothermophilus*, liquid chromatography-mass spectrometry, high resolution electrospray MS

## Abstract

Sixteen strains of five genera of thermophilic bacteria, i.e., *Alicyclobacillus, Brevibacillus, Geobacillus, Meiothermus*, and *Thermus,* were cultivated at a temperature from 42 to 70 °C. Twelve strains were obtained from the Czech Collection of Microorganisms, while four were directly isolated and identified by 16S rRNA gene sequencing from the hot springs of the world-famous Carlsbad spa (Czech Republic). Polyprenol homologs from C40 to C65 as well as free undecaprenol (C55), undecaprenyl phosphate, and undecaprenyl diphosphate were identified by shotgun analysis and RP-HPLC/MS-ESI^+^ (reverse phase high-performance liquid chromatography–high-resolution positive electrospray ionization mass spectrometry). The limit of detection (50 pM) was determined for individual homologs and free polyprenols and their phosphates. Thus, it has been shown that at least some thermophilic bacteria produce not just the major C55 polyprenol as previously described, but a mixture of homologs.

## 1. Introduction

The Eger Graben, called after the river Eger, is a geographic unit in the Czech Republic. It runs SW–NE along the Erzgebirge (Ore Mountains), is part of the European Cenozoic Rift System, and forms an extensive asymmetric tectonic moat. Its origin is related to the response of Alpine Orogeny in the Bohemian Massif, the manifestation of which was volcanic activity that began in the Late Cretaceous but reached its peak in the Tertiary times. This volcanic activity was related to the formation of thermal springs in the world-famous Carlsbad spa area (Czech Republic).

The “Vřídlo” is the official name for the largest spring of thermal water in Carlsbad spa, which has a temperature of 73 °C and contains gasified thermal water with mineralization typical of its complex deep origin, see Appendix A. An average of 2000 L of thermal water is spewed from “Vřídlo” springs every minute, whose water is 18,000 years old [1]. All springs in Carlsbad spa are characterized by a uniform chemistry and are only partial sub-branches from the “Vřídlo” mainstream, which differ in temperature and, depending on it, in the amount of dissolved carbon dioxide. Microbial life in these waters very often remains largely unexplored. Despite being known worldwide, only one microbiological study has been devoted to Carlsbad’s springs, reporting the isolation and characterization of a bacterium of the genus *Thermus* from the spring “Vřídlo” [2].

Distribution, diversity, and activity of thermophilic (or thermotolerant) microorganisms have been analyzed mostly in high-temperature spring waters [3,4,5]. A number of different definitions have been used to describe microorganisms that grow at high temperatures, but commonly used definitions are based on optimal growth temperature [6]. Thermophiles have an optimal growth temperature in the range of 45 to <80 °C [3]. Bacteria isolated by us directly from hot springs are more like thermotolerant strains. Similarly, Hippchen et al. [7] identified, as early as 1981, bacteria related to *Bacillus acidocaldarius* (now *Alicyclobacillus acidocaldarius* [8]) in soil samples that were pasteurized for 10 min at 80 °C. 

Polyisoprenoid alcohols occur from *Archaea* through bacteria to higher plants and animals. In bacteria, polyprenols serve primarily as specific membrane-bound carriers of glycan biosynthetic pathways responsible for the production of cellular structures such as *N*-linked protein glycans and bacterial peptidoglycans. In Gram-negative bacteria, the major accumulated forms are undecaprenyl phosphate (C55-P) and undecaprenyl diphosphate (C55-PP), but not free undecaprenol (C55-OH). In contrast, free undecaprenol is commonly identified in Gram-positive bacteria in addition to the two phosphates [9]. Molecules of polyprenols consist of repeating isoprenoid units linked head-tail with various structural modifications. These include *cis-trans* (E–Z) configurations of double bonds, their saturation (e.g., dolichols) and of course the number of repeating units. Plant polyprenols show the highest diversity in chain length [10] while, with exceptions, only polyprenol C55 (bactoprenol) has been identified in bacteria.

Many published articles [11,12,13] state that *“An interesting feature of polyprenols is their occurrence as the homologous mixtures. Unique in this context are bacteria possessing a single polyprenol (undecaprenol)”*. A complete exception are cyanobacteria, some of which have been shown to produce C35–C45 polyprenols [14]. *Streptococcus mutans* could be a certain exception, but evidence of homologs of polyprenols was only based by gas chromatography on the identity of retention times with standards [15]. Furthermore, decaprenyl-phosphate-arabinose and octahydro-heptaprenyl-phosphate-arabinose have been identified in *Mycobacterium smegmatis* [16]. *Alicyclobacillus acidocaldarius*, formerly *Bacillus acidocaldarius*, is a species of acidophilic, thermophilic Gram-positive bacterium [8]. De Rosa et al. [17] describe a TLC band with retention factor 0.35 which, according to MS, shows a series of molecular ions corresponding to a mixture of C45–C60 polyprenols. From the above overview of already published papers, it is clear that a number of polyprenol homologs also occur in some bacteria, which was confirmed by the data presented in the Results.

The analysis of polyprenols has been described many times, whether in reviews [10,12] or more recently in various articles, see below. Basically, the only method currently used is LC-MS with soft ionization techniques, e.g., electrospray ionization (ESI) [18], atmospheric pressure photoionization [19] or atmospheric pressure chemical ionization [20]. 

High-performance liquid chromatography (HPLC) combined with electrospray ionization tandem mass spectrometry (ESI-MS/MS) is one of the most suitable methods for analyzing and determining the structure of both free polyprenols and their phosphorylated derivatives. This paper presents two examples of the analysis of polyprenols and their derivatives in thermophilic bacteria, either obtained from the collection of microorganisms (CCM-Brno, Brno, Czech Republic) or directly isolated from the hot springs. The first is the use of shotgun analysis, which has several advantages over HPLC-ESI-MS/MS. In particular, it is a fast method which makes it possible to analyze a sample in the order of minutes; another advantage of shotgun analysis over LC-MS analysis is that during direct infusion the mass spectrum displays protonated molecules at constant concentrations, allowing the scanning of precursor-ion scans and/or neutral loss scans. Conversely, the use of LC-MS analysis allows the separation of individual homologs, in this case polyprenols, especially when using reverse phase HPLC (RP-HPLC). Further analysis by tandem MS, thus, allows the exact structure of each of the polyprenol homologs to be determined. This paper focused on the analysis of homologs of polyprenols obtained by saponification of total lipids of sixteen strains of thermophilic bacteria, whether obtained from the collection or directly isolated from the hot springs of the world-famous spa, Carlsbad (Czech Republic).

## 2. Materials and Methods

### 2.1. Chemicals and Standards

Undecaprenol, undecaprenyl-phospate diammonium salt, and undecaprenyl-diphosphate triammonium salt were purchased from Larodan (Malmö, Sweden). All other chemicals were purchased from Merck (Darmstadt, Germany).

### 2.2. Isolation of Thermophilic Bacteria from Hot Springs

Samples were collected from four thermal springs (Štěpánka, Sadový, Mlýnský, and Vřídlo) in Carlsbad, Czech Republic. Sterile 50 mL Falcon tubes were used for the sampling. Each tube was either empty or contained 1.5 mL of sterile yeast extract-peptone-dextrose broth (YPD, in g/L: dextrose 20, peptone 20, yeast extract 10, pH 7.0). The spring water was filled to 15 mL in order to achieve a 10-fold dilution of the media. The samples were then transported back to laboratory in thermos flasks. The Falcon tubes were incubated on rotary shakers (150 rpm) at temperatures matching the temperatures of the given springs, i.e., 58 °C (Štěpánka, Mlýnský and Vřídlo) or 42 °C (Sadový). After 30 days of cultivation in Falcon tubes, each sample was diluted to achieve a final cell concentration of 10^3^ cells/mL. A 0.1 mL aliquot from each of the diluted sample was inoculated to the surface of the agar medium. The agar media used in this study were prepared from the respective micro filtered spring water. The agar media were: Reasoner’s 2A (R2A, HiMedia, Brno, Czech Republic) agar and Thermus 162 agar (HiMedia, Brno, Czech Republic). The plates were incubated at temperatures of the springs (58 °C or 42 °C) for 1–3 weeks.

Several distinct colonies with different morphologies were taken from the above-mentioned agar plates to plate count agar (PCA; HiMedia, Brno, Czech Republic) plates and incubated at 58 °C or 42 °C for 48 h for further isolation and purification. 

### 2.3. Molecular Identification of Four Strains of Thermophilic Bacteria from Hot Springs

Molecular identification was done by sequencing 16S rRNA gene amplified by polymerase chain reaction (PCR) using 16S rRNA gene primers Fwd27 and Rev1492 [21]. PCR amplicons were purified using a High Pure PCR Product Purification Kit (Roche, Basel, Switzerland) following the manufacturer’s protocol. The PCR amplicons were sequenced on an ABI PRISM 3130xl Genetic Analyzer (Applied Biosystems, Waltham, MA, USA). The sequences so obtained were edited by Chromas Lite software (Technelysium Pty Ltd., Brisbane, Australia) and assembled using SeqMan (DNASTAR, Inc., Madison, WI, USA). Searching for 16S rRNA gene sequence similarity was performed at the GenBank data library using the BLASTN program (NCBI, Bethesda, MD, USA).

### 2.4. Cultivation of Four Strains of Thermophilic Bacteria

For the pre-inoculum, four bacterial isolates from hot springs (designated *Geobacillus stearothermophilus* ST-YPD, *Brevibacillus agri* SA-1, *Geobacillus kaustophilus* ML-1, and *Geobacillus stearothermophilus* VR-1) were cultivated in Luria-Bertani broth (LB; 1% tryptone, 0.5% yeast extract, 1% NaCl, pH 7.0) on orbital shaker (150 rpm) for 72 h. For the inoculum, 100 mL of LB was inoculated with 10 mL of pre-culture to a final concentration of OD_600_ 0.2 and incubated on a rotary shaker at 150 rpm for 24 h. For lipid production, 200 mL of LB was inoculated with 10 mL of pre-culture to a final concentration of OD_600_ 0.2 and incubated on an orbital shaker at 150 rpm for 48 h to the stationary phase. The cultivation temperatures for isolate *Geobacillus stearothermophilus* ST-YPD was 42 °C and for the remaining three isolates 58 °C (corresponding to the original spring temperature). After cultivation, the cells were centrifuged (10,000× *g*, 10 min, 4 °C) and washed twice. Biomass yield was determined as dry weight after lyophilization. Cell mass was frozen at −70 °C and lyophilized. An overview of the cultivated bacterial isolates is given in Table 1.

### 2.5. Extraction and Isolation of Polyprenols

The extraction procedure was based on the method of Bligh and Dyer [22]. Briefly, the lyophilized cells (approximately 10 mg) were suspended in a chloroform-methanol mixture (2:1) (~1 mL) for 30 min at 20 °C with stirring, after which, chloroform and water were added and the insoluble material was separated by centrifugation. The aqueous phase was aspirated off and the chloroform phase was evaporated to dryness under reduced pressure. Part of extract was used for shotgun mass spectrometry and another part for saponification.

This latter part of extract was heated to 95 °C for 1 h with 3 mL of 15 M KOH aqueous solution. Non-saponifiable lipids were then extracted three times with hexane, the combined extracts were evaporated to dryness under a stream of nitrogen and analyzed by LC-MS.

The lyophilized cells of the previously cultured strains [23] listed in Table 2 were extracted and further analyzed in the same manner as the four strains mentioned in Table 1.

### 2.6. Analysis of Polyprenols by Shotgun Mass Spectrometry

An LTQ-Orbitrap Velos mass spectrometer (Thermo Fisher Scientific, San Jose, CA, USA) equipped with a heated electrospray interface (HESI) was operated in positive and negative ionization mode. The MS scan range was performed in the FT cell and recorded within a window between 150 and 1500 Da. The mass resolution was set to 100.000, and the ion spray voltage was set at 3.4 kV (in the positive ionization mode) and −2.6 kV (in the negative ionization mode). Both ionization modes used the following parameters: sheath gas flow, 19 arbitrary units (AU); auxiliary gas flow, 8 AU; ion source temperature, 270 °C; capillary temperature, 240 °C; capillary voltage, 55 V; and tube lens voltage, 165 V. Helium was used as a collision gas for collision induced dissociation (CID) experiments. The CID normalization energy of 33% was used for the fragmentation of parent ions. Flow Injection Analysis was used for sample introduction into the heated ESI-MS (H-ESI-MS) ion source. Hexane/propan-2-ol (50/50 *v*/*v*) was used at the flow rate of 125 μL/min. The *m/z* value of the molecular weight-related ion of C55 polyprenol was measured at 765.6917 by FT mode and the mass accuracy of C55 polyprenol was +0.2 ppm, compared with the theoretical value of 765.6919 (Appendix A). The differences between the measured and calculated values for all analyzed compounds, whether in positive or negative ESI, do not exceed 0.3 ppm (0.0003 Da).

### 2.7. LC-MS Analysis of Saponified Polyprenols

The HPLC equipment consisted of a 1090 Win system, PV5 ternary pump and automatic injector (HP 1090 series, Hewlett Packard, Palo Alto, CA, USA), and Ascentis Express column HIRPB-250AM (Hichrom Limited, Berkshire, UK), 250 × 2.1 mm ID, 5 μm particle size. LC was performed at a flow rate of 350 μL/min, with a linear gradient from mobile phase containing propan-2-ol/methanol/aqueous 1 mM ammonium acetate/lithium acetate in methanol (1 µM/mL) (50:40:10:10, *v*/*v*/*v*/*v*) to hexane/propan-2-ol/methanol/aqueous 1 mM ammonium acetate/lithium acetate in methanol (1 µM/mL) (10:70:10:10, *v*/*v*/*v*/*v*)/ with 30 min and held for 10 min. The composition was returned to the initial conditions over 10 min. The whole HPLC flow was introduced into the ESI source without any splitting. The performance was measured by undecaprenol as an internal standard under the conditions given above.

### 2.8. Calibration

The polyprenol standard (commercially obtained C55 polyprenol from Larodan (Malmö, Sweden)) was analyzed separately and SIM at *m/z* 773.7146 ([M+Li]^+^) of C55 polyprenol was obtained. A calibration curve was generated using the area of peak of the SIM corresponding to the different concentrations of standard (1 pM, 10 pM, 50 pM, 100 pM, 1 nM, 10 nM, 100 nM, and 1 μM) and the linear regression of the calibration curve, as well as the signal/noise (S/N) ratio was calculated. 

### 2.9. Statistical Analysis

The statistical analysis was performed using the IBM SPSS Statistics (Statistical Package for the Social Sciences; IBM Corp, 2013) Statistics software, version 26 (IBM^®^ Corporation, Armonk, NY, USA).

## 3. Results and Discussion

### 3.1. Characterization and Identification of Bacterial Isolates

The molecular classification of four strains (SA-1, ML-1, ST-YPD, and VR-1) was carried out by the 16S rRNA gene sequence analysis of an approximately 1.5 kb 16S rRNA fragment amplified from the total DNA of these isolates. The BLAST search showed that these strains belong to the order *Bacillales*, namely to the genus *Brevibacillus* or *Geobacillus.* The partial nucleotide sequence of 16S rRNA gene of these strains (about 1355 nucleotides) showed the maximum nucleotide identity (over 99%) with the strains *Brevibacillus agri* DSM 6348, *Geobacillus kaustophilus* BGSC 90A1, and *Geobacillus stearothermophilus* BGSC 9A20. The conventional physiological and biochemical characterization was performed at the Czech Collection of Microorganisms, Masaryk University, Brno. All strains were defined as Gram-positive, rod-shaped in singlets (or in pairs), aerobic and thermophilic organisms, and spore-forming bacteria (ellipsoidal or oval spores are located at a terminal or subterminal position). The temperature ranges for growth: 40.0–75.0 °C with the optimum 42–58 °C; pH range for growth: 5.0–10.0 with the optimum of 6.0–7.0. On nutrient R2A, the isolate SA-1 formed white, flat colonies with irregular borders. The isolates ML-1, ST-YPD, and VR-1 were characterized by circular umbonate colonies of light orange color with transparent margins. Positive results for the strains ML-1, ST-YPD, and VR-1: catalase, hydrolysis of aesculin, and DNA, nitrite reduction, acids from glucose, mannitol, cellobiose, and fructose. Negative results: hydrolysis of gelatin starch, casein, Tween 80, tyrosine, production of acetoin, acids from xylose, lactose and inositol, growth in Simmons citrate, and arginine dihydrolase. Moreover, the comparison of the two strains ST-YPD and VR-1 was performed; they differed in hydrolysis of o-nitrophenyl-β-galactoside (positive result for the strain VR-1), growth in the presence of 10% NaCl and acids from xylose and mannitol (positive results for the strain ST-YPD). Phenotypic characterization of the strain SA-1 revealed that the isolate tested positive for catalase, urease, hydrolysis of aesculin and casein, growth in glucose, indole, and H_2_S production. The isolate was also capable of forming ellipsoidal spores in swollen sporangia. The isolate tested negative for oxidase, nitrite reduction, Methyl Red/Voges-Proskauer (MR/VP) test, and citrate utilization. Since the results of the molecular analysis were consistent with the phenotypic traits, we designated strains as the *Brevibacillus agri* SA-1, *Geobacillus kaustophilus* ML-1, *Geobacillus stearothermophilus* ST-YPD, and *Geobacillus stearothermophilus* VR-1.

### 3.2. Shotgun Analysis of Polyprenols

A shotgun analysis display of polyprenol phosphates is shown in Figure 1. Using a precursor ion scan at *m/z* 163.0165 (the structure of this ion is shown in the upper right corner of Figure 1), it was possible to obtain preliminary information on homologs of polyprenol phosphates. This image shows only those strains that originate from hot springs, regardless of whether they were spring isolates or strains from the CCM collection. In all strains we identified, albeit sometimes as very minor, homologs were from C40 to C65. Values for [M–H]^−^ both measured and calculated, including summary formulas are given in Appendix A. This table shows a clear agreement between the measured and calculated values for [M–H]^−^ ions, which does not exceed 0.3 ppm (0.0003 Da). Full confirmation of the structure was performed by tandem MS in negative mode for all six polyprenol phosphates. A description of the tandem MS for the lowest identified C40-P is given as an example. A homologous series was identified from the ion at *m/z* 163.0164 (C_5_H_8_O_4_P^−^) to the ion at *m/z* 571.3919 (C_35_H_48_O_4_P^−^, for the structure see Appendix A). These values are in good agreement with already published data [24].

To exclude other homologs (e.g., partially hydrogenated dolichol phosphates, i.e., C40-P, C45-P, and C50-P), that were identified, e.g., in the thermoacidophilic archaeon *Sulfolobus acidocaldarius* [25], we used the shotgun analysis in which both shorter and more saturated compounds would appear in the mass spectra.

### 3.3. LC-MS Analysis of Saponified Polyprenols

Based on these preliminary analyses, see Figure 1, the total lipids were saponified with KOH, see Materials and Methods and unsaponifiable mixture was extracted with hexane and used for LC-MS analysis. Based on both previous good experience [26,27,28] and published data [18], the analyses were performed in positive high-resolution ESI with the addition of Li^+^ ions. The use of SIM for [M+Li]^+^ ions made it possible to identify individual polyprenol homologs. The results of the SIM analysis are shown in Table 2. Appendix A lists the detected and theoretical masses of [M+Li]^+^ ions used for SIM analysis. The data from this table confirm that the total lipid extract of each of 16 strains showed the presence of more than one major homologue of C55 (undecaprenol).

Figure 2 shows that six homologs have been identified in *G. stearothermophilus* VR-1, and their abundances are shown in Table 2. This table also shows the results of analysis of all other strains. The tandem MS of three homologs is showed on Figure 3, i.e., the shortest, longest, and most abundant homologs that have been identified. Tandem MS are relatively simple, the base peak is always an [M+Li]^+^ type ion (e.g., 909.8401 Da), and an abundant ion [M+Li-H_2_O]^+^ (e.g., 891.8297 Da) is also present. Furthermore, abundant ions of the type [M+Li-H_2_O-C_5_H_8_]^+^ were obtained. Polyprenols, due to the double bond in the allylic position to the hydroxyl, very easily lose water to form a conjugate system, see the insertion structure in Figure 3. This formation is facilitated by the transient formation of a six-member ring in which the H atom is transferred to the OH group. In contrast to D’Alexandri et al. [18], where ion formation [M+Li-CH_2_O]^+^ of *m/z* 743.7040 was observed, this ion was not found in polyprenol spectrum, see Figure 3.

The calibration curve was determined based on various concentrations of commercially available C55 polyprenol, see Materials and Methods. The detection limit, i.e., 50 pM (38.6 pg/μL) (defined as the concentration of the sample with an S/N ratio > 3) was obtained. This value is fully consistent with previously published concentrations; e.g., Skorupinska Tudek et al. [29] indicate an interval of 40 pg to 10 ng of the injected standard, or e.g., D’Alexandri et al. [18] where 0.10 nM C55 polyprenol is given. 

As described above, polyprenol homologs other than C55 have been only rarely identified. However, there are several references that shorter and longer homologs than C55 exist. In particular, homologues from C35 to C45 have been identified in the above-mentioned cyanobacteria [14]. These include *A. acidocaldarius* [15] or *S. mutans* [17], where C45–C60 polyprenols have been found. Additionally, the presence of decaprenyl and/or nonaprenyl (C50 and C45) phosphate [30] was proved by mass spectrometry in three genera of Gram-positive bacteria of the genus *Romboutsia*. Four homologs of polyprenols (C45–C60) have been identified in *Streptococcus faecalis,* C55 being predominant [31].

C50-P and trace C45-P were also identified. This again confirmed that although C55 is the major homolog, lower and higher homologs and those polyprenol phosphates are present. Polyprenols of three isolates, *Romboutsia lituseburensis*, *R. ilealis*, and *Romboutsia* sp. were determined and decaprenyl-phosphate (C50) was found to be the major one by mass spectrometry. Furthermore, the presence of decaprenyl phosphate and/or nonaprenyl phosphate as minor polyprenols has been demonstrated in all three species of bacteria of the genus *Romboutsia* [30].

Table 2 shows that in all 16 strains lower and higher homologs than C55 were identified in percent of majority polyprenol (C55). Prenols with less than 8 and more than 13 units were not detected, indicating that its concentration could be below the detection limit (LOD) for ESI^+^-MS (50 pM). For this reason, we cannot exclude the presence of other homologues in trace amounts.

Detection of undecaprenol (C55), undecaprenyl phosphate (C55-P), and undecaprenyl diphosphate (sometimes also called pyrophosphate) (C55-PP) by shotgun negative ESI was first performed on commercially obtained standards. In the tandem MS spectra, a major ion of the type [M–H]^−^ was always identified at *m/z* 765.6919 (C_55_H_89_O^−^), 845.6582 (C_55_H_90_O_4_P^−^), and 925.6246 (C_55_H_91_O_7_P_2_^−^). To fully confirm the structure of the above polyprenols, a tandem electrospray MS in negative mode was performed. In undecaprenyl phosphate, the ion at *m/z* 78.9591 (PO_3_^−^) was identified as the majority, while in undecaprenyl diphosphate the two ions at *m/z* 78.9591 and the ion at 158.9254 (HP_2_O_6_^−^) were identified. An ion of this type has previously been identified, for example, in Chhonker et al. [32] in tandem MS of geranylgeranyl diphosphate. Unfortunately, commercially available standards contained about 5% lower and higher homologs, as evidenced by the negative ESI spectra. For example, two ions with *m/z* 777.5956 (C50-P) and *m/z* 913.7208 (C60-P), respectively, have been identified in undecaprenyl phosphate.

Biosynthesis of polyprenols in Gram-positive bacteria proceeds by successive head-to-tail additions of IPP (isopentenyl diphosphate) that generate longer-chain polyisoprenoid dophosphates. Both C55-P and/or C55-PP are loaded with glycans for the polymerization of cell wall components and can also be replenished during bacterial cell wall synthesis. The individual C55 compounds (C55-OH, C55-P, and C55-PP) can be converted into each other [33]. Based on shotgun analysis and by calibrating three commercially available standards, the ratio between them was determined (Table 3). This ratio is very similar to the published data [34], where polyprenols were found in *Staphylococcus aureus* ratio 113:43:70 nM/mg (dry weight) of cells, i.e., C55-PP:C55-P:C55-OH. Direct determination, e.g., by HPLC, is difficult because, due to the very different polarity, the individual compounds have different retention times [34].

## 4. Conclusions

All the above results of analyses of both sixteen strains and published analyses provide a conclusive evidence that at least thermophilic bacteria biosynthesize different homologs of polyprenols. In addition, various peptides catalyzing reactions in homologs of varying length, i.e., geranylgeranylglyceryl/heptaprenylglyceryl phosphate synthase (ATA58721), farnesyl diphosphate synthase (ALA71315), and *trans*-hexaprenyltranstransferase (ATA60441) are known due to the known sequence of the entire *G. stearothermophilus* genome (NZ_CP008934.1).

## Figures and Tables

**Figure 1 microorganisms-09-01168-f001:**
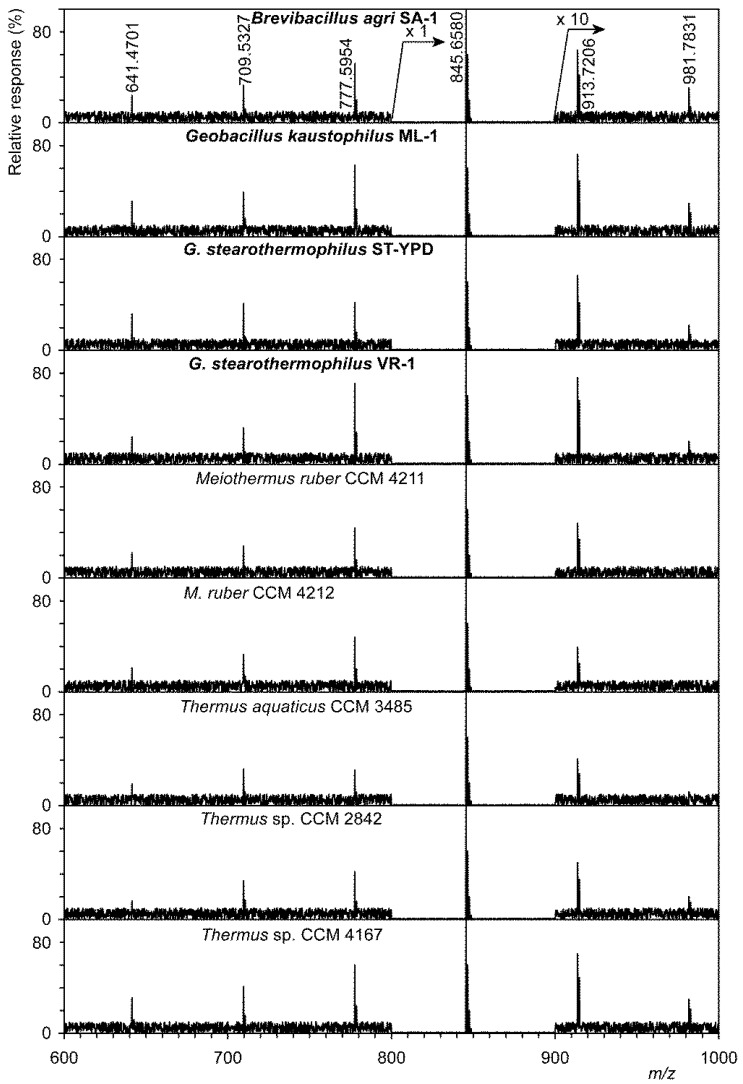
Shotgun analysis of a mixture of polyprenol phosphates from thermophilic bacteria obtained from the Czech Collection of Microorganisms (Brno, Czech Republic) and isolated from hot springs (marked in bold). Precursor ion scan at *m/z* 163.0165, see structure formula at the top right.

**Figure 2 microorganisms-09-01168-f002:**
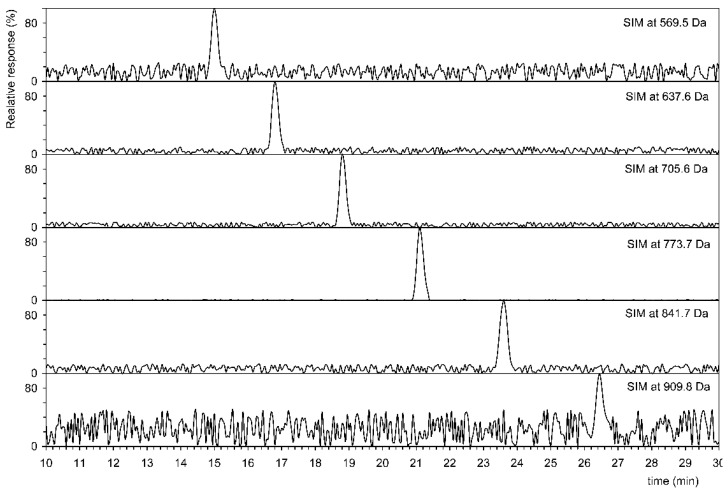
Selected ion monitoring (SIM) indicative of six homologs of polyprenols from C40 (569.5 Da) to C65 (909.8 Da) in *Geobacillus stearothermophilus* VR-1.

**Figure 3 microorganisms-09-01168-f003:**
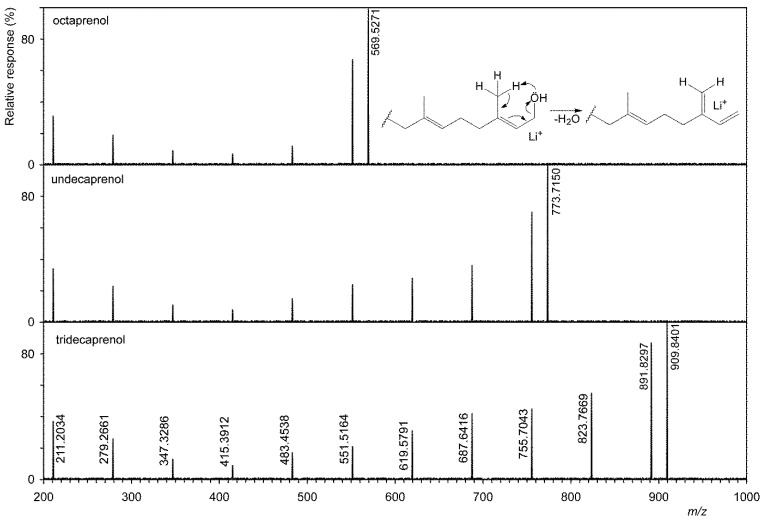
Tandem mass spectra of shortest (octaprenol, i.e., C40), most abundant (undecaprenol, i.e., C55), and longest (tridecaprenol, i.e., C65) polyprenol homologues.

**Table 1 microorganisms-09-01168-t001:** List of the cultivated bacterial isolates from four springs of Carlsbad (Karlovy Vary), Czech Republic.

Isolate	Hot Spring	Cultivation Temperature (°C)	GenBank Accession Numbers
*Brevibacillus agri* SA-1	Sadový	42	MT251434
*Geobacillus kaustophilus* ML-1	Mlýnský	58	MT251494
*Geobacillus stearothermophilus* ST-YPD	Štěpánka	58	MT251887
*Geobacillus stearothermophilus* VR-1	Vřídlo	58	MT251886

**Table 2 microorganisms-09-01168-t002:** Content (relative %) of polyprenol homologs from 16 strains of thermophilic bacteria obtained after saponification using reverse-phase liquid chromatography-positive electrospray mass spectrometry.

Strain	C40	C45	C50	C55	C60	C65
*Alicyclobacillus acidoterrestris* CCM ^a^ 4659 (Apple-grape-raspberry juice) ^b^	0.9 ± 0.2 ^c^	1.8 ± 0.5	5.3 ± 1.5	87.6 ± 2.1	3.5 ± 1.0	0.9 ± 0.2
*Alicyclobacillus acidoterrestris* CCM 4660 (Apple-grape-raspberry juice)	0.0 ± 0.0	1.8 ± 0.7	4.6 ± 1.7	90.9 ± 1.8	2.7 ± 0.9	0.0 ± 0.1
***Brevibacillus agri* SA-1 (spring Sadový) ^d^**	1.7 ± 0.3	2.5 ± 1.2	4.2 ± 0.9	84.8 ± 3.2	5.1 ± 0.6	1.7 ± 0.3
***Geobacillus kaustophilus* ML-1 (spring Mlýnský) ^d^**	2.4 ± 0.3	3.3 ± 1.8	5.1 ± 1.6	81.1 ± 1.4	5.7 ± 1.7	2.4 ± 0.4
*Geobacillus stearothermophilus* CCM 2062 (no information about source)	0.9 ± 0.4	2.6 ± 0.9	3.5 ± 1.0	87.0 ± 2.1	4.3 ± 2.1	1.7 ± 0.4
*Geobacillus stearothermophilus* CCM 5965 (Evaporated milk)	0.9 ± 0.2	0.9 ± 0.5	2.7 ± 0.8	89.2 ± 1.7	3.6 ± 1.6	2.7 ± 0.3
***Geobacillus stearothermophilus* ST-YPD (spring Štěpánka) ^d^**	2.5 ± 0.6	3.4 ± 1.4	3.4 ± 1.4	84.0 ± 3.4	5.0 ± 1.4	1.7 ± 0.5
***Geobacillus stearothermophilus* VR-1 (spring Vřídlo) ^d^**	1.6 ± 0.5	2.5 ± 1.0	5.7 ± 1.6	82.0 ± 2.8	6.6 ± 2.1	1.6 ± 0.4
*Geobacillus thermoglucosidasius* CCM 3731 (soil, Japan, Shimogamo, Kyoto)	0.9 ± 0.5	1.8 ± 0.8	3.5 ± 0.8	88.5 ± 2.2	4.4 ± 1.2	0.9 ± 0.2
*Geobacillus thermoglucosidasius* CCM 3732 (soil, Japan, Shimogamo, Kyoto)	0.9 ± 0.2	1.8 ± 0.4	2.7 ± 0.9	90.1 ± 1.9	3.6 ± 1.3	0.9 ± 0.5
*Meiothermus ruber* CCM 4211 (Thermal pools, Hveragherti, Iceland)	1.8 ± 0.4	2.6 ± 1.6	3.5 ± 1.4	87.7 ± 5.8	4.4 ± 1.8	0.0 ± 0.0
*Meiothermus ruber* CCM 4212 (Thermal pools, Hveragherti, Iceland)	1.7 ± 0.7	2.6 ± 1.1	4.3 ± 2.0	87.0 ± 2.6	3.5 ± 1.1	0.9 ± 0.4
*Thermus aquaticus* CCM 3485 (received from R. A. D. Williams collection, strain DI)	0.0 ± 0.0	0.9 ± 0.4	2.8 ± 0.7	91.7 ± 3.1	2.8 ± 0.9	1.8 ± 0.3
*Thermus aquaticus* CCM 3488 (source: Thermally polluted river near Brussels, Belgium)	1.8 ± 0.3	2.7 ± 0.9	2.7 ± 0.9	88.4 ± 2.9	3.5 ± 0.8	0.9 ± 0.2
*Thermus* sp. CCM 2842 (Hot spring of Kamchatka, Kamchatka, Russia)	0.0 ± 0.0	0.9 ± 0.3	3.6 ± 1.3	89.2 ± 1.7	4.5 ± 2.3	1.8 ± 0.6
*Thermus* sp. CCM 4167 (Hot spring Vřídlo, Carlsbad, Czech Republic)	2.4 ± 0.5	3.3 ± 1.3	4.9 ± 1.2	81.3 ± 3.6	5.7 ± 1.7	2.4 ± 0.4

^a^ Czech Collection of Microorganisms (CCM), Brno, Czech Republic. ^b^ The source(s) from which the strains were isolated according to data from the CCM collection. ^c^ Mean ± S.D. of three independent analyses. ^d^ Isolated bacteria in this study are marked in bold, see Table 1.

**Table 3 microorganisms-09-01168-t003:** Content (relative %) of polyprenol derivatives from 16 strains of thermophilic bacteria obtained after saponification using reverse-phase liquid chromatography-positive electrospray mass spectrometry, see Materials and Methods.

Strain (Source)	C55-OH	C55-P	C55-PP
*Alicyclobacillus acidoterrestris* CCM ^a^ 4659 (apple-grape-raspberry juice)	41 ± 4	17 ± 3	46 ± 7
*Alicyclobacillus acidoterrestris* CCM 4660 (apple-grape-raspberry juice)	36 ± 4	14 ± 6	53 ± 6
***Brevibacillus agri* SA-1 (spring Sadový)**	34 ± 7	27 ± 4	52 ± 8
***Geobacillus kaustophilus* ML-1 (spring Mlýnský)**	44 ± 8	31 ± 7	41 ± 11
*Geobacillus stearothermophilus* CCM 2062 (no information about source)	41 ± 5	24 ± 3	35 ± 7
*Geobacillus stearothermophilus* CCM 5965 (evaporated milk)	50 ± 8	19 ± 5	34 ± 8
***Geobacillus stearothermophilus* ST-YPD (spring Štěpánka)**	43 ± 6	25 ± 9	46 ± 10
***Geobacillus stearothermophilus* VR-1 (spring Vřídlo)**	42 ± 10	48 ± 7	44 ± 9
*Geobacillus thermoglucosidasius* CCM 3731 (soil, Japan, Shimogamo, Kyoto)	36 ± 4	12 ± 3	49 ± 7
*Geobacillus thermoglucosidasius* CCM 3732 (soil, Japan, Shimogamo, Kyoto)	51 ± 4	14 ± 5	32 ± 6
*Meiothermus ruber* CCM 4211 (thermal pools, Hveragherti, Iceland)	46 ± 7	17 ± 2	31 ± 5
*Meiothermus ruber* CCM 4212 (thermal pools, Hveragherti, Iceland)	49 ± 8	16 ± 4	37 ± 4
*Thermus aquaticus* CCM 3485 (received from R. A. D. Williams collection, strain DI)	57 ± 10	19 ± 3	28 ± 8
*Thermus aquaticus* CCM 3488 (source: Thermally polluted river near Brussels, Belgium)	57 ± 7	20 ± 6	25 ± 5
*Thermus* sp. CCM 2842 (hot spring of Kamchatka, Kamchatka, Russia)	57 ± 4	15 ± 5	26 ± 5
*Thermus* sp. CCM 4167 (hot spring Vřídlo, Carlsbad, Czech Republic)	48 ± 3	11 ± 5	24 ± 5

^a^ Mean ± S.D. of three independent analyses. Values in nM/mg lyophilized cells were obtained by means of the calibrating three commercially available standards, see Materials and Methods. Isolated strains in this study are marked in bold.

## Data Availability

The obtained sequences see Table 1 were submitted in GenBank.

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
