# Peer review of "Identification of Homologous Polyprenols from Thermophilic Bacteria"

_microorganisms, 2021, doi:10.3390/microorganisms9061168_

Round 1

Reviewer 1 Report

The manuscript "Identification of homologous polyprenols from thermophilic bacteria" describes a high-resolution mass spectrometric approach for the characterization of polyisoprenoids and polyisoprenoid phosphates. For the characterization of these compounds, an established MS approach using Li+ was chosen, which allows charging the species as M+Li ion in positive ion mode. In general, the manuscript does not provide innovation in terms of methodology. However, the project is well structured and fits the scope of the journal. Before publication, however, there are still minor issues to be addressed:

1) In the shotgun approach, an extremely wide m/z range was used (50-1500). A huge amount of low-molecular-weight contaminants could have affected the results. Moreover, it seems extremely unlikely that the instrumentation could have used such a wide range (usually it is recommended not to exceed a magnitude of molecular weights, e.g., 50-500 or 150-1500)

2) When Li+ is employed, there is usually a greater contribution of contaminants, which are scarcely ionized in standard condition but could more efficiently be lithiated. Were suppression and matrix effects evaluated?

3) I would also strongly recommend a general English revision, especially in the discussion section, e.g., row 300-306.

Author Response

Comments and Suggestions for Authors

The manuscript "Identification of homologous polyprenols from thermophilic bacteria" describes a high-resolution mass spectrometric approach for the characterization of polyisoprenoids and polyisoprenoid phosphates. For the characterization of these compounds, an established MS approach using Li+ was chosen, which allows charging the species as M+Li ion in positive ion mode. In general, the manuscript does not provide innovation in terms of methodology. However, the project is well structured and fits the scope of the journal. Before publication, however, there are still minor issues to be addressed:

1) In the shotgun approach, an extremely wide m/z range was used (50-1500). A huge amount of low-molecular-weight contaminants could have affected the results. Moreover, it seems extremely unlikely that the instrumentation could have used such a wide range (usually it is recommended not to exceed a magnitude of molecular weights, e.g., 50-500 or 150-1500)

We apologize for the mistake in the manuscript, the right range should be 150-1500, as stated in the corrected manuscript.

2) When Li+ is employed, there is usually a greater contribution of contaminants, which are scarcely ionized in standard condition but could more efficiently be lithiated. Were suppression and matrix effects evaluated?

The effects of suppression and matrix were not evaluated because we used selected ion monitoring and precursor ion scan methods, where, on the contrary, we were satisfied that many compounds that could hardly be ionized in the standard state were ionized using Li ion. It did not matter that several thousand ions were identified, see Supplement 2.

3) I would also strongly recommend a general English revision, especially in the discussion section, e.g., row 300-306.

The English revision was performed by a native speaker..

Reviewer 2 Report

Identification of Homologous Polyprenols from Thermophilic Bacteria”

This manuscript describes an effort to identify the polyprenol homologues from C40 to C65 in sixteen thermophilic bacteria strains obtained from the thermal springs and CCM collectionusing  shotgun mass spectrometry and LC-MS analysis. The new information regarding the synthesis of polyprenol shorter or longer than C55 in bacteria are interesting to researchers in this field, however, the data are somewhat preliminary. The manuscript is hard to follow, the Results section is tedious to read and some data require clarification.

Major comments:

1. To generate a calibration curve the Authors used C55 polyprenol only. In my opinion the phosphorylated derivatives should absolutely be used to generate a calibration curve for the quantitative analysis of the mono- and diphosphate polyprenols. Moreover, to clearly confirm the identity of the identified compounds (C40-C65), it seems necessary to compare their retention times with the retention times of the appropriate well defined standards. Such standards are commercially available.

2. Figure 1 – the MS spectra presented in this figure are unreadable. Please show good quality spectra of shotgun analysis. Signals from polyprenol phosphates other than C55-P are invisible. The figure shows the m/z values corresponding to the theoretical masses of [M+Li]+ of C40-C65 polyprenol phosphates. It is surprising that the detected m/z values of the analyzed ions were identical to the theoretical ones. Please indicate appropriate m/z values of ions in the lipid mixture extracted from thermophilic bacteria (such data for one selected microorganism should be provided in the Supplement).

Please provide the name of the chemical compound represented by the structure formula shown on Figure 1 or correct the structure.

3. Figure 2 – The scale on the Y axis (relative response) indicates that signals corresponding to C40-C65 polyprenols have the same intensity. It would suggest that the content of these compounds in the tested samples is the same. It is not consistent with data present in Table 3 which show that C55 polyprenol is the most abundant, while the remaining polyprenols are present in trace amounts. The levels of background, especially for signals corresponding to ions 569.5 Da and 909.8 Da, clearly document that the scale on the Y axis should be corrected.

4. Figure 3 – The MS/MS analyses show a very nice and pure fragmentation pattern of polyprenols. Our experience indicates that it is almost impossible to obtain such an ideal fragmentation pattern from a biological samples. How was this analysis performed? How to explain that the intensity of fragmentation signals are the same for C55 polyprenol as well as for C40 and C65 which content is much lower than C55 in the polyprenol mixture extracted from studied bacteria. Please indicate which signals correspond to [M+Li] and [M+Li-H20] respectively. The theoretical masses of [M+Li] and [M+Li-H20] ions are shown on this figure while instead experimental values should be shown.

5. Table 3. C55-OH, C55-P and C55-PP are not polyprenol homologues. C55-P and C55-PP are phosphorylated derivatives of polyprenol C55. Please correct this. Have all of these compounds been identified by LC-ESI-MS method? There are no data available for this analysis in the manuscript body. Moreover, no data on C55-PP analysis are included . Please show the data (MS spectra) documenting the content of C55-PP in the studied samples. It is unclear how the ratio of polyprenol C55 and its phosphorylated derivatives was calculated. What is more, the sum of the relative content of these compounds (expressed as percentages) exceeds 100%. Please explain in the text and correct the manuscript. .

The authors wrote that they used three commercial standards for LC-ESI-MS analysis. What kind of standards was it? Please complete this information in Materials and Methods section.

6. Figure S1. Why the Authors show the structure of the ion corresponding to C35 polyprenol phosphate if no such compound is identified in the studied bacteria strains? Furthermore, this structure does not describe polyprenol molecule typically isolated from either bacteria or plant tissue. Either correct the structure or provide NMR data documenting the structure presented. Finally, why only (-1) charge is ascribed to the phosphate group?

7. In Conclusions the Authors suggest that “These results are supported by molecular analysis , see GenBank search results. Polyprenyl synthetase family protein has been identified in Geobacillus kaustophilus (WP_044730878), G. stearothermophilus (RLP90700), or Brevibacillus agri (WP_025843728)”. The Authors should explain this in the manuscript, or alternatively remove this statement. Following aspect should be commented in this context.

Polyprenol homologues, from C40 to C65, identified in the thermophilic bacteria do not show a Gaussian-like distribution of homologues typical for polyisoprenoids from Eukaryota. Therefore this observation raises a question whether these polyprenols are produced by the same enzyme that synthesizes bactoprenol (C55) or they are rather products of other enzymes that have not been characterized so far?

Is it possible to identify their  isomery – trans/cis or all-trans? This information addresses the concept of their biosynthetic origin.

8. Moreover, this paper needs revision by the Native speaker. I did struggle to understand several paragraphs. Some sentences are misleading.

l. 259 “To exclude other homologues (e.g. partially hydrogenated dolichol phosphates, i.e. C40-P, C45-P and C50-P), which for example were identified in the thermoacidophilic archaeon Sulfolobus acidocaldarius [25], the shotgun analysis in which both shorter and more saturated compounds would appear in the mass spectra was used.” This sentence has to be rewritten.

l. 342 “Biosynthesis of polyprenols in Gram-positive bacteria proceeds by condensation of pyrosphates to form C55-PP” – in fact successive head-to-tail additions of IPP (isopentenyl diphosphate) generate longer-chain polyisoprenoid diphosphates. This sentence has to be rewritten. As it is now it do not provide enough information about polyprenols biosynthesis.

Minor comment:

Please use polyprenol diphosphates rather than polyprenol pyrosphates.

Author Response

Comments and Suggestions for Authors

“Identification of Homologous Polyprenols from Thermophilic Bacteria”

This manuscript describes an effort to identify the polyprenol homologues from C40 to C65 in sixteen thermophilic bacteria strains obtained from the thermal springs and CCM collectionusing  shotgun mass spectrometry and LC-MS analysis. The new information regarding the synthesis of polyprenol shorter or longer than C55 in bacteria are interesting to researchers in this field, however, the data are somewhat preliminary. The manuscript is hard to follow, the Results section is tedious to read and some data require clarification.

Major comments:

  1. To generate a calibration curve the Authors used C55 polyprenol only. In my opinion the phosphorylated derivatives should absolutely be used to generate a calibration curve for the quantitative analysis of the mono- and diphosphate polyprenols. Moreover, to clearly confirm the identity of the identified compounds (C40-C65), it seems necessary to compare their retention times with the retention times of the appropriate well defined standards. Such standards are commercially available.

We agree with the reviewer that calibration curves for mono- and di-phosphates should be used, but the main focus of the manuscript was not to quantify, but rather refute the dogma that bacteria biosynthesize only undecaprenol. Therefore, we quantified only polyprenol and not their derivatives. Finally, it was found that the bacteria, at least all 16 strains that we analyzed, contain lower and higher homologues of polyprenols, thus confirming our hypothesis, see also below.

I agree with the reviewer that homologues of polyprenols are commercially available (e.g. Larodan), but it is not possible to compare single data, i.e. the retention time that would be obtained by standard chromatography with tandem MS, where data on dozens of ions are compared. Therefore, we consider this analysis to be superfluous.

  1. Figure 1 – the MS spectra presented in this figure are unreadable. Please show good quality spectra of shotgun analysis. Signals from polyprenol phosphates other than C55-P are invisible.

The Figure 1 has been corrected; now the mass spectra are shown on which the noise abundance differs. The spectra are normalized so that the base peak always has a relative intensity of 100%. To determine the abundance, see Table 2.

The figure shows the m/z values corresponding to the theoretical masses of [M+Li]+ of C40-C65 polyprenol phosphates. It is surprising that the detected m/z values of the analyzed ions were identical to the theoretical ones.

We are very sorry, but as we state in our subchapter 3.2. Shotgun analysis of polyprenols, ESI- shotgun was used, therefore, no [M+Li]+ ions can be identified in the mass spectra, so even the figure does not show the theoretical masses of [M+Li]+ ions.

We apologize, the column values have been amended, see also Table S2 for the correct values.

Please indicate appropriate m/z values of ions in the lipid mixture extracted from thermophilic bacteria (such data for one selected microorganism should be provided in the Supplement).

Although we do not consider the addition of such data to be beneficial to the manuscript, nevertheless on reviewer’s request we present a table, see Supplement 2. It is 69 pages of A3 format.

Please provide the name of the chemical compound represented by the structure formula shown on Figure 1 or correct the structure.

The figure does not show a chemical compound, but the structure of the ion, which is correct, see also the paper Analytical Biochemistry 396 (2010) 133-138, where the following statement is reported: "As expected, the entire fragment-ion series starting at m/z 162.7 (C5H8PO4-, 163.02 Da calculated, Fig. 1D) ... ". We measured the value 163.0165 and the theoretical value for C5H8O4P- is 163.0166 Da, which corresponds to the value 163.02 Da after rounding, see above. The structure of this ion is now shown in Fig. S1.

  1. Figure 2 – The scale on the Y axis (relative response) indicates that signals corresponding to C40-C65 polyprenols have the same intensity. It would suggest that the content of these compounds in the tested samples is the same. It is not consistent with data present in Table 3 which show that C55 polyprenol is the most abundant, while the remaining polyprenols are present in trace amounts. The levels of background, especially for signals corresponding to ions 569.5 Da and 909.8 Da, clearly document that the scale on the Y axis should be corrected.

We are very sorry, but this is not entirely true, the signals do not have the same intensity. The reviewer does not take into account noise abundance. In the case of mass spectra it is quite common that the base peak has 100 % abundance. This is also our case. If we displayed this figure, as the reviewer would like, the peaks would be as small as they were in the original Fig. 1.

  1. Figure 3 – The MS/MS analyses show a very nice and pure fragmentation pattern of polyprenols. Our experience indicates that it is almost impossible to obtain such an ideal fragmentation pattern from a biological samples. How was this analysis performed?

We are very sorry, but we have a different opinion. The reviewer’s claim regarding figure 3 applies only to mass spectra, not to tandem mass spectra. In the case of tandem MS, only one peak is analyzed, or better said, the interval +-0.2 ppm from the value of the given ion. The ions have very nice and pure fragmentation tandem MS because there is no other ion in this interval, see Supplements.  

How to explain that the intensity of fragmentation signals are the same for C55 polyprenol as well as for C40 and C65 which content is much lower than C55 in the polyprenol mixture extracted from studied bacteria.

Again, this is relative intensity. When analyzing tandem MS using the Fourier transform (FT) mode, the noise is random, while the signal (ion) is not random, see also the original manuscript “The m/z value of the molecular weight-related ion of C55 polyprenol was measured at 765.6917 by FT mode and the mass accuracy of C55 polyprenol was +0.2 ppm, compared with the theoretical value of 765.6919”.

Please indicate which signals correspond to [M+Li] and [M+Li-H2O] respectively.

Signals correspond to [M+Li]+ and [M+Li-H2O]+ are given in the text, for example m/z 909.8401 versus 891.8297.

The theoretical masses of [M+Li] and [M+Li-H20] ions are shown on this figure while instead experimental values should be shown.

We apologize for this, it was corrected.

  1. Table 3. C55-OH, C55-P and C55-PP are not polyprenol homologues. C55-P and C55-PP are phosphorylated derivatives of polyprenol C55. Please correct this.

The table was corrected.

Have all of these compounds been identified by LC-ESI-MS method? There are no data available for this analysis in the manuscript body.

No, see the text in the original manuscript “Detection of undecaprenol (C55), undecaprenyl phosphate (C55-P), and undecaprenyl diphosphate (sometimes also called pyrophosphate) (C55-PP) by shotgun negative ESI was first performed on commercially obtained standards.”

Moreover, no data on C55-PP analysis are included. Please show the data (MS spectra) documenting the content of C55-PP in the studied samples.

The data of tandem mass spectrum of C55-PP is now in the Supplements (Table S3).

It is unclear how the ratio of polyprenol C55 and its phosphorylated derivatives was calculated. What is more, the sum of the relative content of these compounds (expressed as percentages) exceeds 100%. Please explain in the text and correct the manuscript.

We apologize to the reviewer, the table has been corrected/ rewritten.

The authors wrote that they used three commercial standards for LC-ESI-MS analysis. What kind of standards was it?

It was already mentioned in the original manuscript in the chapter 2.1, see: Undecaprenol, undecaprenyl-phospate diammonium salt, and undecaprenyl-diphosphate triammonium salt were purchased from Larodan (Malmö, Sweden). All other chemicals were purchased from Merck (Darmstadt, Germany).

  1. Figure S1. Why the Authors show the structure of the ion corresponding to C35 polyprenol phosphate if no such compound is identified in the studied bacteria strains? Furthermore, this structure does not describe polyprenol molecule typically isolated from either bacteria or plant tissue. Either correct the structure or provide NMR data documenting the structure presented. Finally, why only (-1) charge is ascribed to the phosphate group?

Figure S1 was deleted.

  1. In Conclusions the Authors suggest that “These results are supported by molecular analysis, see GenBank search results. Polyprenyl synthetase family protein has been identified in Geobacillus kaustophilus (WP_044730878), G. stearothermophilus (RLP90700), or Brevibacillus agri (WP_025843728)”. The Authors should explain this in the manuscript, or alternatively remove this statement. Following aspect should be commented in this context.

The statement was removed from the manuscript.

Polyprenol homologues, from C40 to C65, identified in the thermophilic bacteria do not show a Gaussian-like distribution of homologues typical for polyisoprenoids from Eukaryota. Therefore this observation raises a question whether these polyprenols are produced by the same enzyme that synthesizes bactoprenol (C55) or they are rather products of other enzymes that have not been characterized so far?

The main task of the manuscript was to refute the dogma that bacteria only biosynthesize undecaprenol, which we succeeded in doing. The reviewer's question is so extensive that it would require several time demanding experiments, and even after their implementation it is not clear whether alternative biosynthetic pathways would be found.

Is it possible to identify their  isomery – trans/cis or all-trans? This information addresses the concept of their biosynthetic origin.

Identification of E/Z isomers is again a task for several months, consisting of cultivation in fermenters with at least a few liters of working volume, preparative HPLC to obtain milligram amounts of pure substances and their analysis by 2D NMR. This work is completely outside the scope of the manuscript, where the main goal was to prove that some bacteria biosynthesized lower and higher homologues of undecaprenol.

  1. Moreover, this paper needs revision by the Native speaker. I did struggle to understand several paragraphs. Some sentences are misleading.

The manuscript was checked by a native speaker, less-than-clear parts have been revised and changed.

  1. 259 “To exclude other homologues (e.g. partially hydrogenated dolichol phosphates, i.e. C40-P, C45-P and C50-P), which for example were identified in the thermoacidophilic archaeon Sulfolobus acidocaldarius [25], the shotgun analysis in which both shorter and more saturated compounds would appear in the mass spectra was used.” This sentence has to be rewritten.

The sentence has been rewritten to add clarity.

  1. 342 “Biosynthesis of polyprenols in Gram-positive bacteria proceeds by condensation of pyrosphates to form C55-PP” – in fact successive head-to-tail additions of IPP (isopentenyl diphosphate) generate longer-chain polyisoprenoid diphosphates. This sentence has to be rewritten. As it is now it do not provide enough information about polyprenols biosynthesis.

With permission, we replaced the original sentence with that suggested by the reviewer which to our mind provides a clearer picture of polyprenol biosyntehsis.

Minor comment:

Please use polyprenol diphosphates rather than polyprenol pyrosphates. Thank you for the comment, the manuscript has been edited accordingly.

Round 2

Reviewer 2 Report

I believe the authors provided a reasonable reply and fixed my major points of concern. Therefore, I suggest for the manuscript to be accepted.